# Intermittent Fasting Alleviates Risk Markers in a Murine Model of Ulcerative Colitis by Modulating the Gut Microbiome and Metabolome

**DOI:** 10.3390/nu14245311

**Published:** 2022-12-14

**Authors:** Jingjing Wu, Da Man, Ding Shi, Wenrui Wu, Shuting Wang, Kaicen Wang, Yating Li, Liya Yang, Xiaoyuan Bian, Qiangqiang Wang, Lanjuan Li

**Affiliations:** 1State Key Laboratory for Diagnosis and Treatment of Infectious Diseases, National Clinical Research Center for Infectious Diseases, National Medical Center for Infectious Diseases, Collaborative Innovation Center for Diagnosis and Treatment of Infectious Diseases, The First Affiliated Hospital, Zhejiang University School of Medicine, Hangzhou 310003, China; 2Division of Hepatobiliary and Pancreatic Surgery, Department of Surgery, The First Affiliated Hospital, Zhejiang University School of Medicine, Hangzhou 310003, China

**Keywords:** intermittent fasting, gut microbiome, ulcerative colitis, metabolome

## Abstract

Clinical trials have demonstrated the health benefits of intermittent fasting (IF). However, the potential mechanism of IF in alleviating dextran sulfate sodium (DSS)-induced colitis is not fully understood. The present study was mainly designed to explore the dynamic changes in the gut microbiota and metabolome after short-term (2 weeks) or long-term (20 weeks) IF and therefore clarify the potential mechanisms by which IF ameliorates DSS-induced colitis in a murine model. Thirty-two C57BL/6 male mice were equally divided into four groups and underwent IF intervention for 2 weeks (SIF group, *n* = 8), 20 weeks (LIF group, *n* = 8), or were allowed free access to food for 2 weeks (SAL group, *n* = 8) or 20 weeks (LAL group, *n* = 8). The thirty-two C57BL/6 male mice were accepted for the diet intervention of 2 weeks of IF or fed ad libitum. Colitis was induced by drinking 2% DSS for 7 days. Our findings showed that short-term IF prominently elevates the abundance of *Bacteroides*, *Muibaculum* and *Akkermansia* (*p* < 0.001, *p* < 0.001, *p* < 0.001, respectively), and decreased the abundance of *Ruminiclostridium* (*p* < 0.05). Long-term IF, however, decreased the abundance of *Akkermansia* and obviously increased the abundance of *Lactobacillus* (*p* < 0.05, *p* < 0.001, respectively). Metabolites mainly associated with nucleoside, carbohydrate, amino acid, bile acid, fatty acid, polyol, steroid and amine metabolism were identified in the faeces using untargeted GC/MS. In particular, inosine was extremely enriched after short-term IF and long-term IF (*p* < 0.01, *p* < 0.01, respectively); butyrate, 2-methyl butyric acid and valeric acid were significantly decreased after short-term IF (*p* < 0.001, *p* < 0.001, *p* < 0.01, respectively); and 2-methyl butyric acid was significantly increased after long-term IF (*p* < 0.001). The abundance of lithocholic acid (LCA), one of the secondary bile acids, increased significantly after short-term and long-term IF based on UPLC–MS/MS (*p* < 0.001, *p* < 0.5, respectively). Of note, IF markedly mitigated DSS-induced acute colitis symptoms and down-regulated pro-inflammatory cytokines IL-1α, IL-6, keratinocyte-derived chemokine (KC) and G-CSF levels in the serum (*p* < 0.01, *p* < 0.001, *p* < 0.05, *p* < 0.001, respectively). Furthermore, a correlation analysis indicated that the disease activity index (DAI) score and serum levels of IL-1α, IL-6, KC, and G-CSF were negatively correlated with the relative abundance of *Akkermansia* and the faecal metabolites LCA and inosine. This study confirmed that IF altered microbiota and reprogramed metabolism, which was a promising development in the attempt to prevent DSS-induced colitis. Moreover, our findings provide new insights regarding the correlations among the mucosal barrier dysfunction, metabolome, and microbiome.

## 1. Introduction

Data from numerous preclinical studies underscore the negative impact of excessive carbohydrate uptake and favour the beneficial effects of reducing caloric intake [1,2]. Caloric restriction is the only nongenetic intervention that has consistently been found to extend both the mean and maximal lifespan across model organisms [2,3]. Intermittent fasting (IF) encompasses several regimens of fasting, such as alternate day fasting, fast-mimicking diet and time-restricted eating. Research also suggests that IF is superior to other caloric restriction in enhancing long-term memory consolidation and adult hippocampal neurogenesis [4]. However, most individuals have difficulty in maintaining prolonged caloric restriction. IF is much more tolerable and manageable than constant caloric restriction; both trigger similar biological pathways. According to recent studies, it is safe for healthy people and nonobese humans to practice IF for several months [5]. Intermittent fasting has become a point of interest for researchers, as both sleeping and activity patterns have changed during IF treatment. It also impacts the circadian rhythms of various hormones, such as cortisol, insulin, growth hormone, leptin, ghrelin, prolactin, sex hormones and so on [6]. Many studies have shown that IF is associated with protection against oxidative stress, a lower metabolic rate, anti-inflammatory effects, and increased autophagy, antitumor activities, and neuroprotection [1,5]. Moreover, a number of studies have illustrated a shift in the gut microbiota composition caused by IF [7]. However, detailed information regarding dynamic changes in the faecal microbiota and metabolome during IF is elusive. 

Non-pharmacologic therapies like diet intervention have drawn serious attention in recent decades. Epidemiological data suggest that both a vegetarian diet and a gluten-free diet failed to accrue robust evidence on alleviating ulcerative colitis (UC) [8]. Consequently, the low FODMAP (fermentable oligosaccharides, disaccharides, monosaccharides, and polyols) diet (LFD) may be helpful, especially for overlapping functional symptoms in those with inflammatory bowel disease (IBD) [9,10]. However, the data on whether or not LFD impacts IBD disease activity is inconclusive. Fasting/reduced-calorie diets are promising interventions [11]. A study conducted in 2008 found that a significant decrease in a clinical colitis activity index was noted in 60 IBD patients during Ramadan [12]. Since the ability of IF to reduce systemic inflammation has been clarified in prior studies [13], systematic analysis of the influence of IF on the UC is not well known. As generally known, composition of the diet, dietary pattern, and long-term dietary habits shape the diversity and composition of gut microbiome [14]. It is widely accepted that the gut microbiome is a contributor to the development of UC. However, bacterial components and microbial metabolites such as short-chain fatty acids (SCFAs), secondary bile acids, and tryptophan also play a crucial role in the pathogenesis of UC. Most related research in this area has focused on the influence of IF on the gut microbiome. Fewer studies have demonstrated the dynamic changes of both microbiome and metabolome. Therefore, we firstly adopt this present study to explore the effects of strict IF on the gut microbiota and metabolome to fill this gap. We adopted alternate day fasting (ADF), which is defined as strict 24-h periods without caloric intake and then 24-h intervals with ad libitum food consumption. This is the first study to provide a comprehensive analysis of the dynamic alterations of gut microbiota as well as metabolome after short-term (2 weeks) and long-term (20 weeks) IF periods. Secondly, we also generated a mouse colitis model to illustrate the mechanism of IF on DSS-induced colitis.

## 2. Methods

### 2.1. Short-Term IF and Long-Term IF Experimental Design

Thirty-two male specific pathogen-free (SPF) mice were randomly assigned into four groups. Each group contained eight mice. Mice in the short-term IF group (SIF) or long-term IF group (LIF) were fed every other day (food pellets were provided or removed at 9 am each day) for 2 weeks or 20 weeks, while mice in the control group were allowed free access to food (SAL or LAL group). Animals were weighed at 9 am after a feeding day. Mouse faeces (0.18–0.22 g) were collected before sacrifice. At the end of the experiment, all mice were humanely euthanized to collect serum and colon segment for further analysis.

### 2.2. DSS-Induced Colitis and Experimental Design

Thirty-two six-week-old male mice were randomly assigned into one of the following groups (*n* = 8 each): ad libitum-fed group (AL group), intermittent fasting group (IF group), ad libitum-fed group with DSS (ALD group), and intermittent fasting group with DSS (IFD group). After 2 weeks of diet intervention, the ALD group and IFD group were given 2% (*w*/*v*) DSS (molecular weight 36,000–50,000, MP Biomedicals, Santa Ana, CA, USA) dissolved in drinking water for 7 days (Days 0–6). Fresh DSS solution was prepared every 2nd day. The daily intake of DSS was recorded in the Appendix A. The other mice were provided regular drinking water. 

### 2.3. Microbiome Sequencing of Faecal Samples

Faecal samples were collected in the morning after a feeding day at the end of the 2- week or 20-week diet intervention and frozen at −80 °C until DNA extraction. QIAamp Fast DNA Stool Mini Kit (Qiagen, Hilden, Germany) were used to extract DNA from faeces in accordance with the manufacturer’s instructions. The 16S rRNA genes were amplified by V3–V4 region (338F: 50-ACTCCTACGGGAGGCAGCA-30, 806R: 50-GGACTACHVGGGTWTCTAAT-30) and sequenced on the Illumina MiSeq platform (Illumina, San Diego, CA, USA).

### 2.4. Bile Acids Analysis of Faecal Samples

Approximately 10 mg of faecal samples was accurately weighed. Then, approximately 25 mg of precooled grinding beads and 200 µL of acetonitrile/methanol (*v*/*v* = 8:2) mixed solvent containing 10 µL internal standard were added. The mixture was combined in a tissue homogenizer. After centrifugation at 13 and 500 rpm for twenty min at 4 °C, 10 µL of supernatant was diluted with 90 µL of a 1:1 acetonitrile/methanol (80/20) and ultrapure water mixture and mixed by vertexing. After centrifugation at 4 °C for 20 min at 13,500 rpm, the samples were placed in a 4 °C refrigerator for sampling analysis based on the ultra-performance liquid chromatography/tandem mass spectrometry (UPLC–MS/MS) method.

### 2.5. Metabolomic Profiling Analysis of Faecal Samples

Mice faeces were accurately weighed to a portion of 15 mg and 800 µL pre-cooled methanol (Sigma-Aldrich, St. Louis, MO, USA) was added. After being centrifuged and filtered, the supernatant was transferred to a 1.5-mL tube containing 20 µL of 1 mg/mL heptadecanoic acid (Sigma-Aldrich) as the internal standard. The mixture was dried and concentrated with nitrogen (Aosheng, Hangzhou, China). The residue was reconstituted in 50 µL of 15 mg/mL methoxyamine pyridine solution (Sigma-Aldrich) and incubated for 24 h (37 °C). Then, 50 µL of N,O-bis(trimethylsilyl)trifluoroacetamide (BSTFA) with 1% trimethylchlorosilane (TMCS; Sigma-Aldrich) was added, and the solution was incubated again for 2 h (70 °C) for derivatization. Metabolomic analysis was performed with a gas chromatography-mass spectrometry (GC-MS on an Agilent 7890A GC system coupled to an Agilent 5975C inert mass selective detector (MSD system (Agilent Technologies, Santa Clara, CA, USA.

The data were analyzed using Qualitative Analysis B.07.00 (Agilent, Santa Clara, CA, USA). Metabolites were identified using the NIST 17 database. Principal component analysis (PCA) and orthogonal partial least-squares-discriminant analysis (OPLS-DA) were performed to visualize metabolic differences among the experimental groups. A VIP value in the OPLS-DA model of >1 and the *p* values < 0.05 were taken as the standard criteria to measure the influence of metabolites for sample classification. The KEGG database and the Human Metabolome Database (HMDB) were used to search for metabolic pathways associated with the characteristic metabolites.

### 2.6. SCFAs Quantification

Briefly, 20 mg of faeces were mixed with 500 µL internal standard (hexanoic acid-d3, 10 µg/mL). After homogenization and centrifugation (15,000 rpm, 5 min, 4 °C), the supernatant was transferred into an Eppendorf tube with 5% concentrated sulfuric acid (supernatant vs. 5% concentrated sulfuric acid, volume 10:1). Then, an equivalent volume of ethyl acetate was added to the mixture. After centrifugation and incubation at 4 °C for 30 min, the supernatant was transferred to chromatographic vials equipped with 150 µL inserts before the gas chromatography mass spectrometry (GC-MS) analysis. The analysis was performed on an Agilent 7890A GC oven coupled to an Agilent 5975C inert mass selective detector (Agilent Technologies, Santa Clara, CA, USA). SCFAs were identified by the times of retention in accordance with the standard solutions. Standard mixes of 0.01, 0.1, 1, 10, 100, 1000 µL/mL and blank were also prepared in the same way as the samples.

### 2.7. Histopathological Evaluation and Immunofluorescence Analyses

Tissue sections were collected and soaked in 10% neutral buffered formalin for 24 h at room temperature. The paraffin-embedded samples were cut into 5-μm thickness and stained with haematoxylin and eosin (H&E) and finally observed under a light microscope for further histopathological analysis.

Similarly, paraffin-embedded colon sections were stained for Zonula occludens-1 (ZO-1) (tight junction proteins) (Proteintech, Rosemont, IL, USA) with standard immunofluorescence staining procedures as previously detailed.

### 2.8. Serum Cytokine Analysis

Plasma cytokine levels were quantified with a Bio-Plex Pro mouse cytokine 23-plex assay kit (BioRad, Hercules, CA, USA) according to the manufacturer’s protocol.

### 2.9. Statistical Analysis

The data are shown as means and standard error of means. Whether the data satisfied the normal distribution criteria was assessed by the Kolmogorov–Smirnov test. Most data, such as body weight, gut length, adipose tissue weights, and SCFAs content, was analysed by ANOVA followed by post hoc least significant difference (LSD) testing. The nonparametric Mann–Whitney U test (α = 0.05) was used to compare differences in the abundance of bile acids among these groups. Statistical significance was defined as *p* < 0.05.

## 3. Results: Short-Term IF and Long-Term IF Experiments

### 3.1. IF Impacts Body Weight and Adipose Tissue Contents

We first investigated the influence of IF on body weight. We found that the body weight of mice did not decrease dramatically until 3 after weeks of intermittent fasting (Figure 1A,B). Studies have shown that the fatty acid metabolism is distinctly impacted after IF [8]. Thus, subcutaneous white adipose tissue (subcutaneous WAT), visceral epididymal WAT, and interscapular brown adipose tissue (interscapular BAT) were weighed. Two weeks of IF significantly lessened the fat mass, especially the visceral epididymal WAT. On the other hand, IF didn’t affect the subcutaneous WAT or even improve the interscapular BAT mass (Figure 1C). Identical results were observed after 20 weeks of IF (Figure 1D). Our findings suggest that IF might accelerate adaptive non-shivering thermogenesis and the burning of fat. Yet, IF treatment had no influence on the length of the small intestine and colon regardless of short-term IF or long-term IF (Figure 1E,F).

### 3.2. Influence of IF on the Gut Microbiota

We surveyed the microbiome composition of the stool samples from these groups. The gut microbiota undergoes extensive changes across the lifespan [9]. In this part, we compared the influence of short term IF on the gut microbiota between the SAL and the SIF groups, and long term IF on the gut microbiota between the LAL and the LIF group. No significant differences of both overall microbial diversity and the community richness have been found after IF (*p* > 0.05, Table 1). Then, we performed a nonmetric multidimensional scaling (NMDS) analysis to investigate the β diversity in gut microbial communities among these groups. For both short-term IF and long-term IF, a distinct separation of gut microbiota among the ad libitum-fed group and fasting group have been found (short-term, stress = 0.063, r^2^ = 0.393, *p* = 0.001; long-term, stress = 0.087, r^2^ = 0.185, *p* = 0.027, Appendix A).

After 2 weeks of IF, the *Firmicutes* abundance substantially decreased (47.4% in the SAL group, 20.7% in the SIF group, *p* < 0.01) and the proportion of *Bacteroidetes* increased (49.0% in the SAL group, 73.8% in the SIF group, *p* < 0.001). However, 20 weeks of IF did not significantly change the dominant phyla *Firmicutes* and *Bacteroidetes*. We observed that *Verrucomicrobia* decreased significantly after IF (short-term, *p* < 0.001; long-term, *p* < 0.05, respectively).

*Muribaculaceae* and *Akkermansiaceae* dramatically increased in the SIF group compared with the SAL group at the family level (*p* < 0.05, *p* < 0.001, respectively). Additionally, the abundance of *Lachnospiraceae* and *Ruminococcaceae* decreased significantly due to 2-week IF (*p* < 0.01, *p* < 0.05, respectively). Interestingly, *Akkermansiaceae* decreased after 20-week of IF while *Lactobacillaceae* clearly increased (*p* < 0.001, *p* < 0.01, respectively).

At the genus level, the abundance of *Bacteroides*, *Muribaculum* and *Akkermansia* was prominently elevated in the SIF group (*p* < 0.001, *p* < 0.001, *p* < 0.001, respectively). *Ruminiclostridium* decreased remarkably in the SIF group (*p* < 0.05), which was not observed in LIF group. However, *Akkermansia* decreased after 4 months of IF while *Lactobacillus* obviously increased obviously (*p* < 0.05, *p* < 0.001, respectively).

We then performed the linear discriminant analysis effect size (LEfSe) analysis to identify differentially abundant biomarkers between the mice from the IF group and the control group. We found that mice subjected to 2-week IF harboured a distinctively higher abundance of the phylum *Bacteroidetes*, which was due to the abundance of *Muribaculaceae* and *Bacteroidaceae* (LDA score (−log10) > 4, Figure 2A)*. Lachnospiraceae* and *Ruminococcaceae* were separately enriched in the SAL group (LDA score (−log10) > 4). In addition, compared with the LAL group, 20-week IF remarkably improved the abundance of *Lactobacillus* (LDA score (−log10) > 4, Figure 2B) and reduced the proportion of *Akkermansia* and *Parasutterella* (LDA score (−log10) > 4).

### 3.3. Bile Acids Profiles of Faeces after IF

We next focused on the bile acids profiles of faeces after IF. After 2 weeks of IF, the relative abundance of the primary bile acids TCDCA and TCA increased remarkably (*p* < 0.01, *p* < 0.01, respectively, Figure 3A). In contrast, the relative abundance of αMCA and CA decreased (*p* < 0.05 and *p* < 0.01, respectively). For secondary bile acids, HDCA, LCA, muroCA and 6-keloLCA increased significantly (*p* < 0.05, *p* < 0.001, *p* < 0.05 and *p* < 0.01, respectively, Figure 3B). However, the relative abundance of TωMCA and ACA decreased (*p* < 0.05 and *p* < 0.01, respectively).

After 20 weeks of IF, the relative abundance of the primary bile acids βMCA, TαMCA, TCDCA and TCA increased significantly (*p* < 0.05, *p* < 0.05, *p* < 0.05 and *p* < 0.05, respectively, Figure 3C), while the relative abundance of αMCA decreased (*p* < 0.001). For the secondary bile acids, the relative abundance of LCA increased dramatically (*p* < 0.05, Figure 3D). However, the abundance of ωMCA and 7-keloLCA decreased (*p* < 0.05 and *p* < 0.01, respectively).

### 3.4. Metabolomic Profiling Analysis after IF

To assess metabolic alterations in response to IF, we applied the untargeted strategy to study the faecal metabolome that was associated with functional features of intestinal microorganisms. A total of 281 metabolites were ultimately identified from the faecal samples.

The metabolic clustering between groups was evaluated by both principal component analysis (PCA) and orthogonal partial least-squares-discriminant analysis (OPLS-DA). The PCA model provides an overview of the cluster tendencies between the SAL and SIF groups and between the LAL and LIF groups (Appendix A). Initially, a significant separation of clusters was exhibited between the SAL and SIF groups (R^2^X = 0.635) and between the LAL and LIF groups (R^2^X = 0.564). Appendix A shows the distinct differences in metabolite spectra induced by short-term IF (R^2^Y = 0.996, Q^2^ = 0.957), and Appendix A depicts the metabolite profiling discrimination according to OPLS-DA after long-term IF treatment (R^2^Y = 0.949, Q^2^ = 0.861).

A total of 80 and 93 important metabolites were selected based on the standard criteria (VIP value > 1, *p* values < 0.05) between the SAL and SIF groups and between the LAL and LIF groups according to the OPLS-DA model (Appendix A). Glucose, kynurenic acid, inosine, 3,4-dihydroxyphenylacetic acid and 5-methyluridine were extremely enriched in the SIF group (Table 2). It was determined that 2,6-Diaminopimelic acid, galactinol, trisaccharide, inulotriose and inosine were extremely enriched in the LIF group. Moreover, 41 metabolites were shared biomarkers among the 2-weeks and 20-weeks of IF, most of which were mainly associated with nucleoside, carbohydrate, amino acid, bile acid, fatty acid, polyol, steroid and amine metabolism. Among them, thirty-one potential biomarkers were enriched in the SIF and LIF groups and nine were depleted in the SIF and LIF groups. Isochlorogenic acid was the sole biomarker that was enriched in the SIF group but depleted in the LIF group.

Moreover, 41 metabolites were shared biomarkers between 2-weeks and 20-weeks of IF, most of which were mainly associated with nucleoside, carbohydrate, amino acid, bile acid, fatty acid, polyol, steroid and amine metabolism. Among them, thirty-one potential biomarkers were enriched in the SIF and LIF groups and nine were depleted in the SIF and LIF groups. Isochlorogenic acid was the sole biomarker that was enriched in the SIF group but depleted in the LIF group.

### 3.5. Influence of IF on SCFAs in the Gut

The concentrations of SCFAs, including acetate, propionate, isobutyric acid, butyrate, 2−methyl butyric acid and valeric acid, were evaluated in the faecal samples after IF. After 2 weeks of IF, the levels of the six kinds of SCFAs decreased in the SIF group. In particular, butyrate, 2-methyl butyric acid and valeric acid were significantly decreased in the SIF group (*p* < 0.001, *p* < 0.001, *p* < 0.01, respectively, Figure 4A). However, 20 weeks of IF increased the levels of most SCFAs except isobutyric acid. The level of 2-methyl butyric acid in the LIF group was extensively enriched, more so than that of the LAL group (*p* < 0.001, Figure 4B).

## 4. IF Alleviates DSS-Induced Colitis

### 4.1. IF Alleviates Colitis Symptoms and Colon Injury after DSS Administration

We assessed colon damage in a chemically DSS induced colitis model, one which exhibits several characteristics resembling human UC. We found that after the 2-week period of IF intervention, no significant difference was found in body weight among the AL, IF, ALD and IFD groups (Figure 5A). At the end of the experiment, reduced body weight loss was observed in the mice in the IFD group. On the other hand, the cumulative food consumption of the four groups was calculated, and the food consumption of the IF group was 295.78 g/100 g body mass (BM), while that of the AL group was 277.21 g/100 g BM (Figure 5B), the IFD group was 292.14 g/100 g BM, and the ALD group was 269.16 g/100 g BM. These data indicate that IF did not affect the total food intake of mice. A reduction in colon length is a marker of intestinal damage after DSS treatment and a phenotypic feature used to assess IBD severity [10]. We assessed the effects of DSS on colon length, and no differences were detected between the AL group and IF group, suggesting that IF had no adverse effects on colon length. Because of DSS drinking, the ALD group had significant reductions in colon length compared with the IFD group (*p* < 0.001, Figure 5C). Inflammatory profiles during the onset of colitis could be indicated by the weight and length of the spleen. Thus, we further assessed the impacts of IF on the weight and length of the spleen. IF did not cause an increase in either the weight or the length of the spleen. However, a significant increase in both the weight and length of the spleen was induced by DSS admission, and the values were much higher in the ALD group (*p* < 0.05, *p* < 0.01, Figure 5D,E). After one week of DSS administration, bloody diarrhoea appeared in the faeces of mice. According to the DAI scoring system, mice in the ALD group presented much higher scores (Figure 5F). Moreover, the ALD group also presented a markedly elevated modified DAI score compared to the IFD group (Figure 5G).

Histology of distal colon segments showed that the intestinal mucosal tissue structure was normal between the AL group and the IF group (Figure 6A,B). The ALD group exhibited serious injuries, including damage of crypts, loss of goblet cells, inflammatory cell infiltration and severe mucosal destruction, which were strikingly abrogated in the IFD group (Figure 6C,D). As shown in Figure 6E, IF protected against DSS-induced colon injury (*p* < 0.05).

To further investigate whether IF could improve mucosal barrier functions, the tight junction protein zonula occludens1 was analysed by means of immunofluorescence staining. Of note, no significant differences were displayed between the AL group and the IF group. The ALD group presented increased tight junction structure defects with destroyed crypts and a disrupted apical region. Intriguingly, ZO-1 was abundantly expressed in the colon of IFD group, resulting in stabilized mucosal integrity (Figure 7).

### 4.2. IF Relieves DSS-Induced Systemic Inflammation

Serum cytokines are crucial factors during the onset of colitis [11]. To explore whether IF could alleviate colitis by regulating inflammation, 23 serum cytokines were assessed in our study. The levels of cytokines measured in the AL group and the IF groups were not significantly different. Consistent with the current reports, there was an increase of pro-inflammatory cytokines IL-1α, IL-1β, IL-2, IL-3, IL-6, IL-9, IL-12P40, IL-12P70, IL-17, IFN-γ, and TNF-α; anti-inflammatory cytokines IL-4 and IL-10; and chemokines KC, MCP-1, MIP-1β, G-CSF, and GM-CSF in the ALD group (Table 3). As expected, the IL-1α, IL-6, KC, and G-CSF contents in the ALD group reached a higher level than that of those in the IFD group (*p* < 0.01, *p* < 0.001, *p* < 0.05, *p* < 0.001, respectively, Figure 8A–D).

### 4.3. Elucidating the Mechanism Responsible for IF

The correlations among the representative microbes, metabolic biomarkers from the SAL and SIF groups with the DAI index and serum pro-inflammatory cytokines after DSS administration were determined using Spearman’s rank correlation analysis (Figure 9). To this end, our data revealed that the DAI score was positively correlated with the serum levels of IL-1α and G-CSF. The relative abundance of *Akkermansia* was negatively correlated with the DAI and serum levels of IL-1α, IL-6, KC, and G-CSF. However, the relative abundance of *Akkermansia* was positively correlated with the metabolites of LCA and inosine in the faeces. LCA and inosine were also negatively correlated with the DAI and serum levels of IL-1α, IL-6, KC, and G-CSF. We also found that the DAI was positively correlated with the relative abundance of *Ruminiclostridium* and negatively correlated with the relative abundance of *Muribaculum*.

## 5. Discussion

Gut microbiota diversity is critical for linking the diet and the host’s physiology and pathology, and is influenced by dietary composition and patterns [12]. Our study systematically demonstrated dynamic alterations in the gut microbiota and metabolome after IF intervention. A research paper published in 2022 has demonstrated that IF was associated with worsening of the partial Mayo score [13]. The study found that serum CRP and stool calprotectin did not show a significant change after IF [13]. On the contrary, another study indicated that IF exhibited protective effects against colitis and related behavioural disorders [14]. Our results also demonstrated that IF reversed the DSS-induced body weight loss, increased DAI score, colon length shortening, serum inflammatory cytokines, and inflammation. Moreover, in our research we found some discriminative biomarkers of gut microbes and microbial metabolites after IF. Studies have shown that the α-diversity (richness) and β-diversity (variety) of gut microbiota was impacted by fasting [15]. According to a study, IF decreased the enrichments of colitis-related microbes such as *Shigella* and *Escherichia* and increased the relative abundance of *Rikenellaceae*, *Lactobacillus*, *Coproccus*, and *Ruminococcus* [13]. It is worth mentioning that in our study IF did not change the α-diversity of gut microbiota. Yet, the composition of gut microbiota in each group was significantly different when analysed by NMDS analysis. The abundance of *Firmicutes* substantially decreased and the abundance of *Bacteroidetes* increased after short-term IF. It has been reported in the previous studies that a higher *Firmicutes* to *Bacteroidetes* ratio is associated with obesity [16]. On the contrary, we also found that long-term IF did not significantly change the dominant phyla *Firmicutes* and *Bacteroidetes*. At the genus level, the abundance of *Bacteroides*, *Muribaculum* and *Akkermansia* was prominently elevated in the SIF group. However, *Akkermansia* decreased after 20 weeks of IF while *Lactobacillus* obviously increased. In the study conducted by Liu et al., IF compared to the control group led to a significant increase in the abundance of *Lactobacillus*, *Ruminococcus*, and *Akkermansia strains* [17]. *Akkermansia* strains have been associated with metabolic benefits such as a reduction in the severity of fatty liver and intestinal inflammation [18]. The intestinal tract is colonized by a large and diverse microbial community that plays a critical role in the maintenance of intestinal homeostasis in UC patients. Previous studies found that the contents of *Faecalibacterium prausnitzii*, *Clostridium clusters IV* and *XIVa*, *Bacteroides*, *Roseburia species*, *Eubacterium rectale*, *Escherichia coli*, *Fusobacterium*, and *Candida albicans* affect the onset and progression of UC [19]. In parallel with these studies, our sequencing data also presented that IF specifically modified faecal microbiota. The relative abundance of *Akkermansia* was markedly elevated after IF, alleviated the DAI score in DSS-induced colitis, and reduced the concentrations of serum cytokines. Additionally, Spearman’s rank correlation analysis suggested that *Akkermansia* has a nonnegligible impact on LCA and inosine metabolism. Moreover, preventive effects of *Akkermansia muciniphila* on UC have been clarified previously [20,21]. The underlying mechanism may be the accelerated development of intestinal stem cell-mediated epithelium [22]. In addition, *Akkermansia muciniphila* activates NLRP3 expression to ameliorate symptoms in DSS-induced colitis [23]. This probiotic was also associated with diminished microbiota encroachment in other diseases [24,25,26]. Our correlation analysis has shown that DAI was positively correlated with *Ruminiclostridium* and negatively correlated with *Muribaculum*. This result concurs with the previous studies that *Ruminiclostridium* was positively correlated with the cytokines, DAI, and the pathological score [27], and *Muribaculum* was regarded as beneficial bacteria in alleviating DSS-induced colitis [28].

To further explore the gut microbiota with their functional states, metabolomics analysis by GC/MS and UPLC-MS/MS was integrated. A total of 41 metabolites were shared biomarkers after short-term and long-term IF, most of which were mainly associated with nucleoside, carbohydrate, amino acid, bile acid, fatty acid, polyol, steroid and amine metabolism. Other crucial factors that are also involved in the maintenance of intestinal homeostasis included intestinal motility and barriers [29]. Analysis of faecal metabolomics revealed that the primary metabolite of adenosine inosine was significantly enriched in response to short-term IF. Inosine and guanosine are obtained mainly by microbial fermentation and used as important foodstuff additives in our daily life [30]. We found that after short-term IF, the content of inosine in the gut and the weight of interscapular brown adipose tissue was significantly elevated. Extracellular inosine has been associated with energy expenditure in brown adipocytes, as demonstrated in recent research [31]. Novel findings show that inosine plays an important role in many physiological and pathophysiological processes [32]. For instance, inosine is therapeutically useful in enhancing the efficacy of immune checkpoint blockade [32,33]. A recent study elegantly demonstrated the preventive effects of inosine on DSS-induced colitis through promoting A_2A_R/PPARγ-dependent mucosal barrier function [34]. Peculiarly, we also observed that short-term IF was attributed to the enrichment of inosine in the lumen, which could partially explain the beneficial effects of IF on UC.

Bile acids also play a critical role in maintaining intestinal homeostasis. They keep a balance between bile acid-metabolizing bacteria and bile acid-sensitive bacteria. Previously, bile acids were considered potent antibacterial compounds. Since bile acids are major regulators of glucose and lipid homeostasis, microbiota-bile acid crosstalk contributes highly to the weight gain in mice after calorie restriction [35]. Studies have demonstrated that secondary bile acids dramatically decreased in DSS-treated mice faeces [36,37]. Our targeted metabolomics analysis of bile acids showed the profiles of both primary and secondary bile acids altered significantly after short-term and long-term IF. Notably, short-term IF significantly increased secondary bile acids HDCA, LCA, muroCA and 6-keloLCA. In long-term IF, the relative abundance of LCA increased dramatically. LCA, DCA, and HDCA in the faeces were found to ameliorate DSS-induced colitis and accelerate mucosal repair in mice [38]. LCA were the prominent feature in the group with short-term IF and long-term IF. As one of the most hydrophobic acids, LCA showed reduced toxicity to bacteria in the caecal microbiome [39]. It is required for homeostatic intestinal epithelial renewal and fate specification and for regeneration after colitis induction. Moreover, LCA has been reported to control host immune responses in the intestinal lamina propria by directly modulating the balance of TH17 and Treg cells [40]. When LCA is connected to intestinal PXR, Toll-like receptor 4 signalling is inhibited and intestinal pro-inflammatory responses are attenuated in the early stages [41]. When LCA is connected to TGR5, intestinal stem cells are activated, and epithelial regeneration is accelerated [42]. In summary, LCA participates in the process of mucosal repair through different mechanisms.

Studies have revealed that microbiota-derived metabolite SCFAs have the ability to influence the mucosal immune system. Firstly, SCFAs can maintain immune homeostasis, including modulation of regulatory B cells [43]. Secondly, SCFAs inhibit the production of IL-17 from γδ T cells in IBD patients [44]. Then, SCFAs promote antimicrobial peptides, such as RegIIIγ and β-defensins, in the regulation of intestinal homeostasis [45]. A 28-day treatment of IF improved levels of acetate, propionate, and butyrate in the faecal samples [46]. Our data demonstrated that short-term IF failed to increase the levels of SCFAs in the faeces. We proposed that the mechanisms by which IF alleviates UC were independent of SCFAs.

## 6. Conclusions

Several studies have reported that IF prompted recovery from colitis in animal models [47,48]. However, the roles of gut microbiota involved in colitis with IF regimen need to be further investigated. Our observations suggest that IF exerts anti-inflammatory effects via modulating the intestinal microbiota composition and increasing the production of microbiota-derived metabolites. IF significantly elevated the relative abundance of both *Muribaculum* and *Akkermansia* in the gut, which attenuated DSS-induced colitis. Moreover, the increased concentrations of inosine and secondary bile acids like LCA produced by the gut microorganisms probably contributed highly to the anti-colitis effects of IF. Taken together, this work suggests that IF induces metabolic reprogramming in colonic tissues in a gut-microbiota-dependent manner. Encouraged by the observed benefits of IF on colitis, our data provides new insights into the potential use of IF as an efficacious supplement in individuals with UC.

## Figures and Tables

**Figure 1 nutrients-14-05311-f001:**
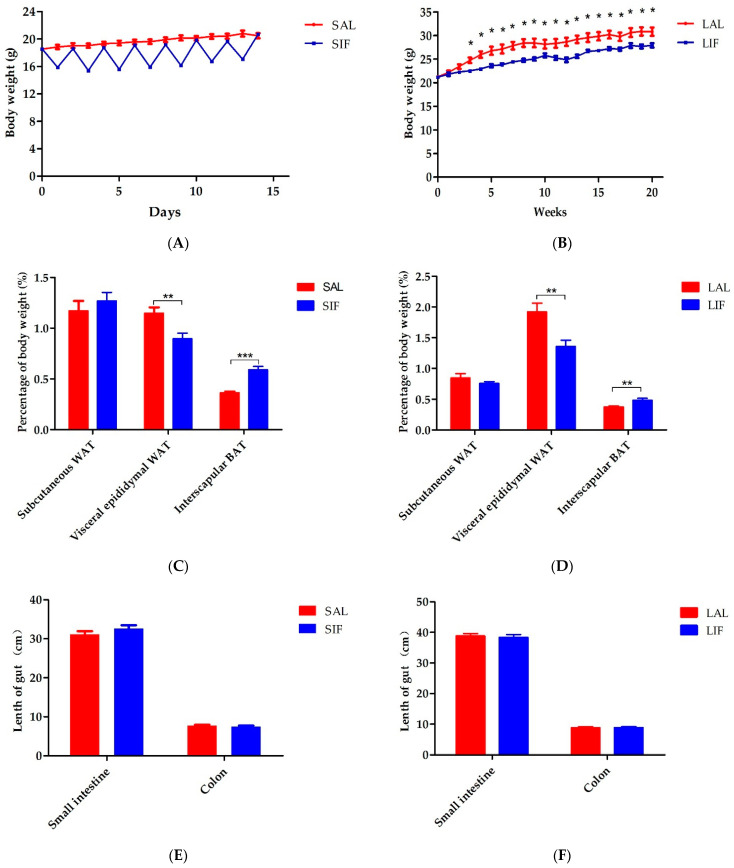
IF impacts the body weight and adipose tissue contents: (**A**) body weight with short-term IF; (**B**) body weight with long-term IF; (**C**) adipose tissue percentage of body weight with short-term IF; (**D**) adipose tissue percentage of body weight with long-term IF; (**E**) length of gut with short-term IF; and (**F**) length of gut with long-term IF. SAL: Short-term of ad libitum-fed group; SIF: Short-term of intermittent fasting group; LAL: Long-term of ad libitum-fed group; LIF: Long-term of intermittent fasting group. * *p* < 0.05, ** *p* < 0.01, *** *p* < 0.001.

**Figure 2 nutrients-14-05311-f002:**
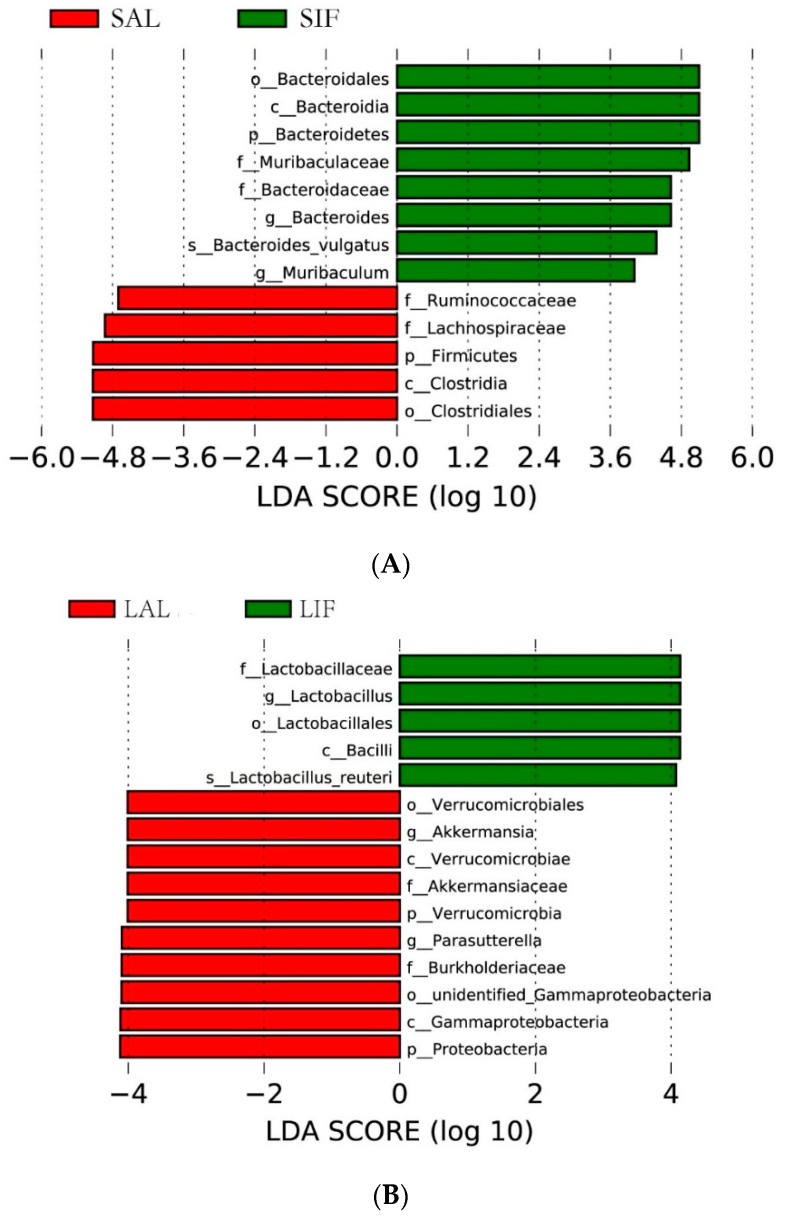
Discriminative biomarkers identified in the populations after IF: (**A**) discriminative biomarkers with an LDA score of >4 in the SAL and SIF groups; and (**B**) discriminative biomarkers with an LDA score of >4 in the LAL and LIF groups. SAL: Short-term of ad libitum-fed group; SIF: Short-term of intermittent fasting group; LAL: Long-term of ad libitum-fed group; LIF: Long-term of intermittent fasting group.

**Figure 3 nutrients-14-05311-f003:**
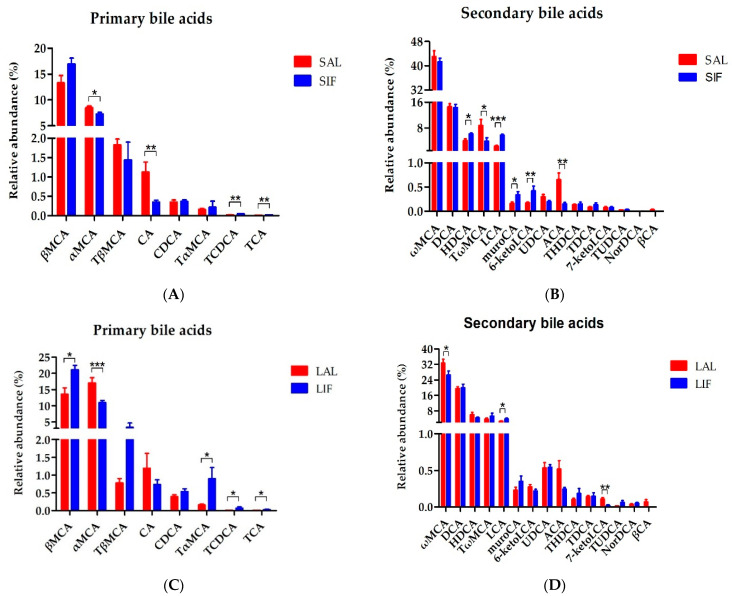
Bile acid profile of faeces after IF: (**A**) primary bile acids in the faeces with short-term IF; (**B**) secondary bile acids in the faeces with short-term IF; (**C**) primary bile acids in the faeces with long-term IF; and (**D**) secondary bile acids in the faeces with long-term IF. SAL: Short-term of ad libitum-fed group; SIF: Short-term of intermittent fasting group; LAL: Long-term of ad libitum-fed group; LIF: Long-term of intermittent fasting group. * *p* < 0.05, ** *p* < 0.01, *** *p* < 0.001.

**Figure 4 nutrients-14-05311-f004:**
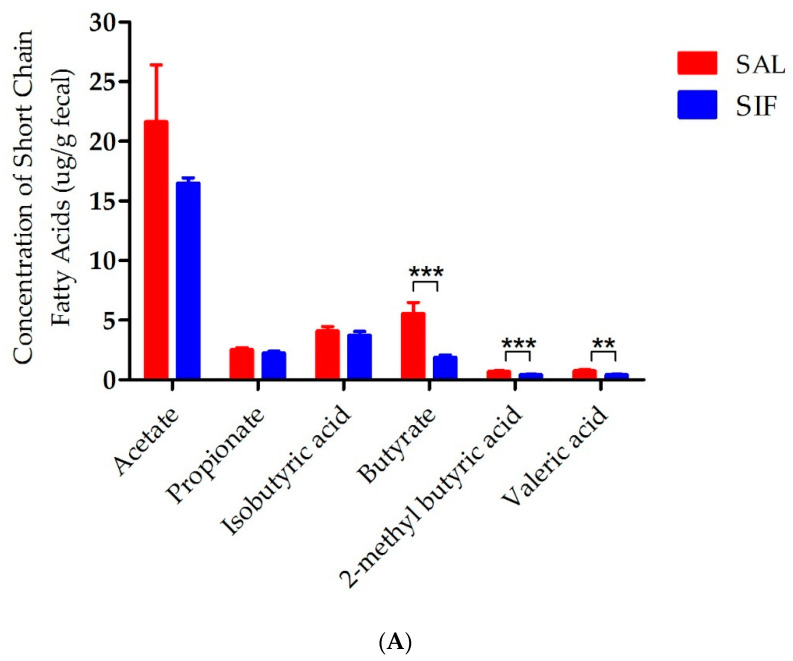
Concentrations of SCFAs in the faeces after IF: (**A**) SCFA concentration in the SAL and SIF groups; and (**B**) SCFA concentration in the LAL and LIF groups. SAL: Short-term of ad libitum-fed group; SIF: Short-term of intermittent fasting group; LAL: Long-term of ad libitum-fed group; LIF: Long-term of intermittent fasting group. *p* < 0.05, ** *p* < 0.01, *** *p* < 0.001.

**Figure 5 nutrients-14-05311-f005:**
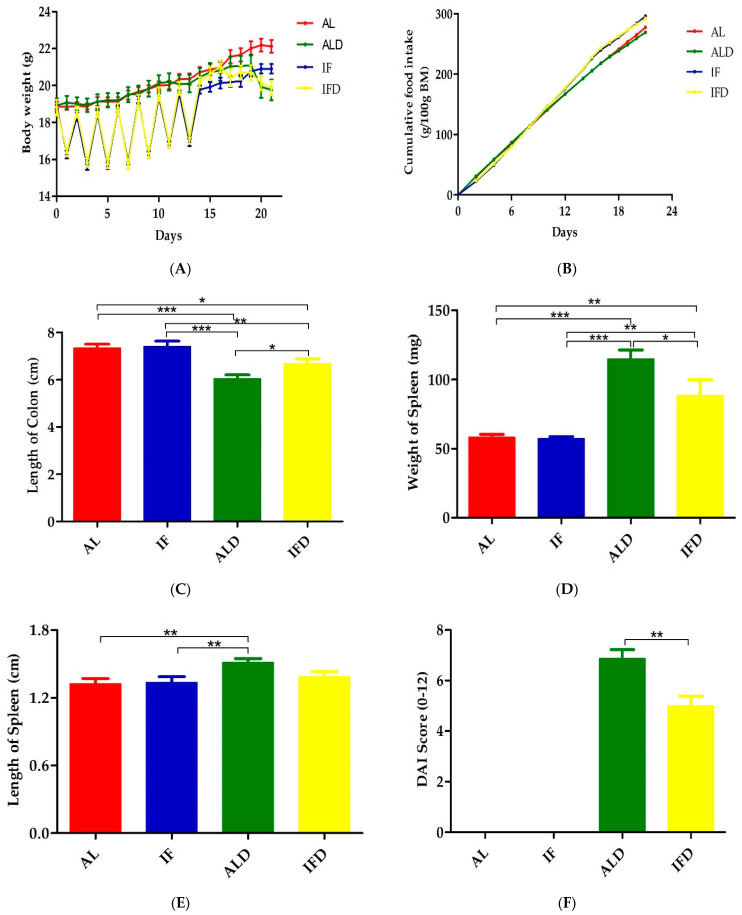
IF alleviates symptoms induced by DSS: (**A**) Body weight; (**B**) Cumulative food intake; (**C**) Colon length; (**D**) Spleen weight; (**E**) Spleen length; (**F**) DAI score; and (**G**) Modified DAI score. AL: ad libitum-fed group; IF: intermittent fasting group; ALD: ad libitum-fed group with DSS; IFD: intermittent fasting group with DSS; DAI: Disease activity index; BM: Body mass. * *p* < 0.05, ** *p* < 0.01, *** *p* < 0.001.

**Figure 6 nutrients-14-05311-f006:**
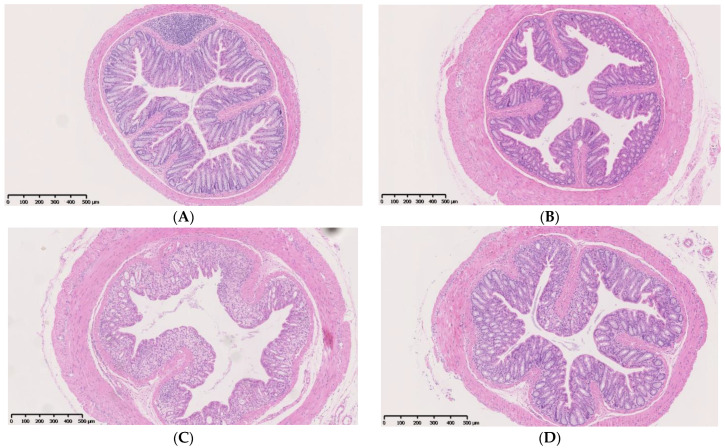
IF ameliorated DSS-induced intestinal mucosal damage: (**A**) representative images of the colon histology in AL group; (**B**) representative images of the colon histology in IF group; (**C**) representative images of the colon histology in ALD group; (**D**) representative images of the colon histology in IFD group; and (**E**) histopathology scores of the four groups. AL: ad libitum-fed group; IF: intermittent fasting group; ALD: ad libitum-fed group with DSS; IFD: intermittent fasting group with DSS. * *p* < 0.05, *** *p* < 0.001.

**Figure 7 nutrients-14-05311-f007:**
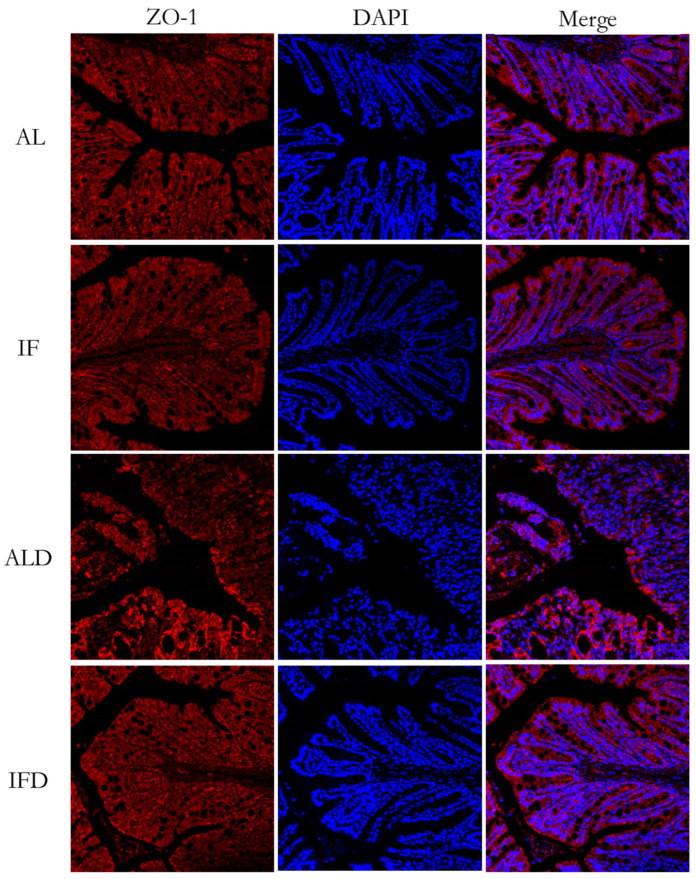
IF alleviates gradual intestinal barrier impairment: Representative colon histology was assessed by ZO-1 immunofluorescence staining. AL: ad libitum-fed group; IF: intermittent fasting group; ALD: ad libitum-fed group with DSS; IFD: intermittent fasting group with DSS.

**Figure 8 nutrients-14-05311-f008:**
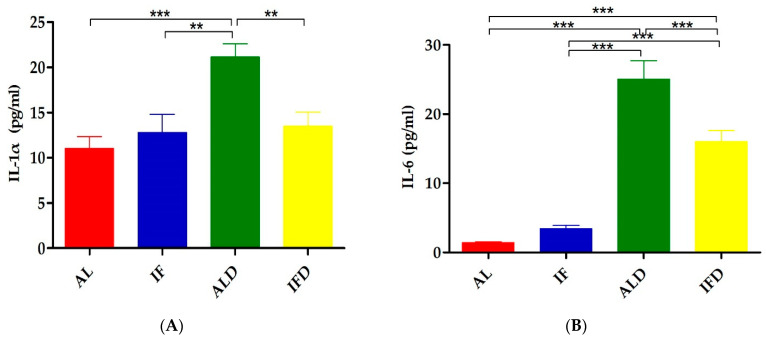
IF alleviates DSS-induced pro-inflammatory cytokines in serum: (**A**) concentration of IL-1α in the serum; (**B**) concentration of IL-6 in the serum; (**C**) concentration of KC in the serum; and (**D**) concentration of G-CSF in the serum. AL: ad libitum-fed group; IF: intermittent fasting group; ALD: ad libitum-fed group with DSS; IFD: intermittent fasting group with DSS. * *p* < 0.05, ** *p* < 0.01, *** *p* < 0.001.

**Figure 9 nutrients-14-05311-f009:**
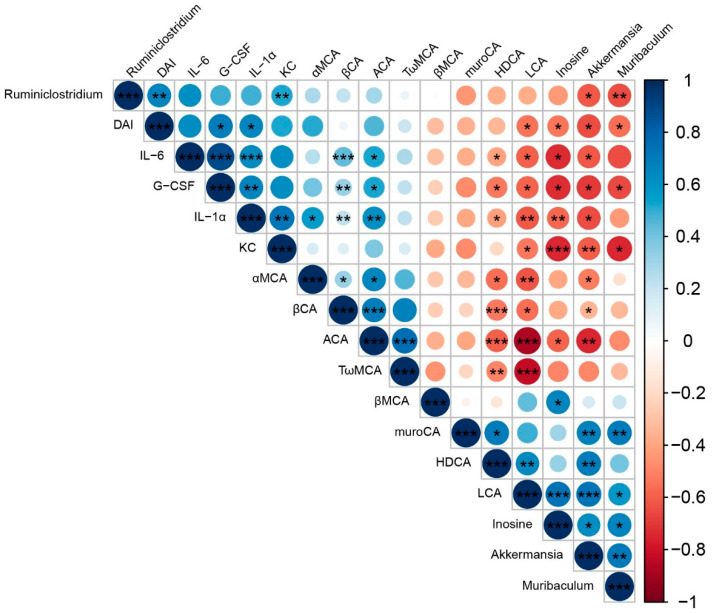
Heat map of the Spearman’s rank correlation analysis of the representative microbes, metabolic biomarkers, DAI score, and inflammatory indexes among the three groups. Color key and circle size indicate the association strength. Dark blue indicates a stronger positive correlation, dark red indicates a stronger negative correlation, and white indicates no correlation. The asterisk in the dot indicates the *p* value, as follows: * *p* < 0.05, ** *p* < 0.01, *** *p* < 0.001.

**Table 1 nutrients-14-05311-t001:** Richness and diversity of the faecal microbiota of each group.

Index	2-Week Intermittent Fasting		20-Week Intermittent Fasting	
	SAL	SIF	LAL	LIF
OTUs	601 ± 191	520 ± 76	230 ± 25	253 ± 20
Ace	756 ± 241	671 ± 145	246 ± 27	269 ± 22
Chao1	741 ± 224	655 ± 146	246 ± 28	268 ± 22
Shannon index	5.56 ± 0.44	5.2 ± 0.28	4.01 ± 0.57	4.26 ± 0.43
Simpson index	0.94 ± 0.02	0.93 ± 0.01	0.86 ± 0.06	0.87 ± 0.05

All data are given as the mean ± SEM. SAL: Short-term of ad libitum-fed group; SIF: Short-term of intermittent fasting group; LAL: Long-term of ad libitum-fed group; LIF: Long-term of intermittent fasting group

**Table 2 nutrients-14-05311-t002:** Effects of IF on the faecal metabolome.

Metabolites		SAL vs. SIF			LAL vs. LIF		
	RT	VIP Value	FC	*p* Value	VIP Value	FC	*p* Value
1,2-ethanediamine	28.91	1.1	1.8	<0.01	1.403	2.18	0.01
2,6-diaminopimelic acid	20.77	2.19	4.96	<0.01	2.498	7.77	<0.01
2-hydroxy-3-methylbutyric acid	8.5	1.39	2.08	<0.01	1.753	3.77	<0.01
2-hydroxycaproic acid	9.87	1.29	1.94	<0.01	1.632	2.93	<0.01
2-indolecarboxylic acid	21.34	1.75	3.44	<0.01	1.051	1.65	<0.01
2-keto-l-gluconate	21.25	1.16	1.72	<0.01	1.346	2.19	<0.01
3,5-dihydroxyphenylglycine	32.04	2.27	6.77	<0.01	1.777	3.28	<0.01
3-epicholic acid	37.42	2.03	0.16	0.01	1.426	0.34	<0.01
5-methyluridine	30.53	2.36	7.09	<0.01	1.833	3.9	<0.01
6-deoxyglucose	19.95	1.64	2.65	<0.01	1.384	2.38	<0.01
6-hydroxy-2-aminohexanoic acid	16.43	1.54	2.55	<0.01	1.048	1.75	0.04
Aspartate	13.59	1.44	0.43	<0.01	1.346	0.13	0.04
Butane-2,3-diol	5.83	1.54	2.48	<0.01	1.13	1.75	<0.01
Butanedioic acid	11.34	1.82	3.14	<0.01	1.086	1.59	0.02
Cellobiose	21.4	1.12	0.49	<0.01	1.008	0.55	<0.01
Cholesterol	28.53	1.37	0.37	<0.01	1.486	0.34	<0.01
Citramalic acid	14.6	1.81	3.16	<0.01	1.077	1.62	0.02
Cyclohexylamine	8.54	1.22	0.35	0.03	1.524	0.25	0.02
D-arabinose	18.82	1.52	2.43	<0.01	1.201	1.98	<0.01
Deoxycholic acid	37.57	1.91	0.17	<0.01	1.733	0.19	0.01
D-tagatose	23.09	2.14	3.4	<0.01	1.337	2.16	<0.01
D-xylose	20.89	1.2	2.66	0.04	1.927	7.62	0.02
Galactaric acid	24.84	1.23	1.97	<0.01	1.172	1.87	0.02
Galactinol	22.86	1.74	4.52	0.03	2.445	9.9	<0.01
Galacto-hexodialdose	22.29	1.53	2.28	<0.01	1.431	3.27	<0.01
Galactose	22.84	1.65	3.63	0.02	2.047	9.5	0.02
Gluconic acid lactone	17.36	1.36	0.39	0.02	1.39	0.35	0.01
Glycolic acid	7.01	1.41	2.01	<0.01	1.293	2.11	<0.01
Inosine	32.3	2.57	11.3	<0.01	2.131	5.85	<0.01
Inulotriose	31.78	2.24	6.31	<0.01	2.133	7.41	<0.01
Isochlorogenic acid	25.34	1.37	2.14	0.01	1.292	0.45	<0.01
Kynurenic acid	31.85	2.61	11.9	<0.01	1.927	3.33	<0.01
L-asparagine	18.92	1.49	0.32	0.02	1.369	0.14	0.03
Leucinic acid	9.76	1.36	1.98	<0.01	1.517	2.78	<0.01
Melibiose	28.06	1.62	2.86	<0.01	2.082	6.05	<0.01
Methyl beta-d-glucopyranoside	23.24	1.64	2.73	<0.01	1.351	2.34	0.02
N-carbamoylaspartate	21.1	1.84	3.21	<0.01	1.622	2.71	<0.01
Phenyllactic acid	17.01	1.24	1.81	<0.01	1.523	2.86	<0.01
*p*-hydroxylphenyllactic acid	14.66	1.74	2.81	<0.01	1.548	2.67	<0.01
Tranexamic acid	14.23	1.24	0.5	<0.01	1.171	0.54	<0.01
Xylofuranose	17.7	1.46	2.72	<0.01	1.901	5.13	<0.01

FC: fold change. SAL: Short-term of ad libitum-fed group; SIF: Short-term of intermittent fasting group; LAL: Long-term of ad libitum-fed group; LIF: Long-term of intermittent fasting group.

**Table 3 nutrients-14-05311-t003:** IF alleviates DSS-induced pro-inflammatory cytokines in serum.

Inflammatory Cytokines (pg/mL)	AL Group	IF Group	ALD Group	IFD Group
IL-1α	11.06 ± 3.63	12.78 ± 5.74	21.13 ± 4.18 ***	13.5 ± 4.37 ##
IL-1β	2.32 ± 1.45	5.27 ± 3.56	12.82 ± 5.66 ***	9.57 ± 2.23
IL-2	5.23 ± 4.96	4.22 ± 1.28	8.74 ± 2.96 *	7.54 ± 2.32
IL-3	3.60 ± 1.11	4.70 ± 1.23	8.14 ± 2.73 ***	7.20 ± 2.51
IL-6	1.41 ± 0.36	3.40 ± 1.47	25.06 ± 7.52 ***	16.04 ± 4.48 ###
IL-9	6.35 ± 1.47	10.78 ± 3.37	15.80 ± 9.10 **	13.25 ± 4.22
IL-12P40	821.25 ± 99.80	1028.30 ± 263.93	1361.20 ± 372.46 **	1354.28 ± 397.44
IL-12P70	119.24 ± 21.53	154.53 ± 52.00	221.96 ± 97.32 **	207.09 ± 81.13
IL-17	125.89 ± 30.09	154.09 ± 34.62	233.67 ± 72.18 ***	254.26 ± 49.73
IFN-γ	12.11 ± 2.17	16.35 ± 4.57	24.72 ± 10.51 **	22.34 ± 6.40
TNF-α	45.89 ± 12.23	60.53 ± 21.1	108.86 ± 44.35 ***	105.80 ± 34.11
IL-4	1.45 ± 1.51	3.73 ± 2.75	4.19 ± 2.68 *	3.85 ± 2.51
IL-10	23.87 ± 4.74	33.39 ± 7.49	54.97 ± 18.95 ***	54.69 ± 16.33
KC	47.62 ± 36.27	51.54 ± 12.63	78.03 ± 14.70 **	49.06 ± 13.50 #
MCP-1	107.15 ± 28.00	180.90 ± 56.60	284.37 ± 63.82 ***	261.27 ± 49.07
MIP-1β	23.30 ± 8.41	24.18 ± 8.21	47.94 ± 9.94 ***	40.57 ± 11.28
G-CSF	166.68 ± 32.68	182.75 ± 30.30	2920.54 ± 1543.24 ***	1111.86 ± 368.53 ###
GM-CSF	20.96 ± 8.65	25.04 ± 9.78	44.82 ± 9.48 ***	37.93 ± 10.15
IL-5	4.55 ± 2.23	8.48 ± 4.43	8.57 ± 3.30	8.78 ± 5.72
Eotaxin	1531.17 ± 240.65	1645.73 ± 282.52	1108.29 ± 388.48 *	1250.41 ± 356.2
MIP-1α	3.41 ± 0.78	3.13 ± 0.71	2.98 ± 0.72	2.62 ± 0.86
RANTES	153.38 ± 27.01	168.62 ± 23.41	133.68 ± 18.17	148.04 ± 24.01

Data are shown as the mean ± SEM. AL: ad libitum-fed group; IF: intermittent fasting group; ALD: ad libitum-fed group with DSS; IFD: intermittent fasting group with DSS. ALD group compared with AL group, * *p* < 0.05, ** *p* < 0.01, *** *p* < 0.001; ALD group compared with IFD group, # *p* < 0.05, ## *p*< 0.01, ### *p* < 0.001.

## Data Availability

The data presented in this study are available on request from the corresponding author.

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
