# Peer review of "Intermittent Fasting Alleviates Risk Markers in a Murine Model of Ulcerative Colitis by Modulating the Gut Microbiome and Metabolome"

_nutrients, 2022, doi:10.3390/nu14245311_

Round 1
Reviewer 1 Report
The study of Jingjing Wu aimed to assess the dynamic of changes in the gut microbiota and metabolome after 2 weeks or 20 weeks IF and explore how IF affects DSS-induced colitis in mice.
Overall, the manuscript is properly written in English and logically organized. But I have a few major and minor suggestions to improve the manuscript.
Introduction The Introduction provides synthetic information about the current knowledge about intermittent fasting (IF) and its impact on general human health. Then, the role of diet and IF in inflammatory bowel diseases was briefly mentioned. Point 1. In my opinion, the Introduction would be enriched by adding a few sentences informing about the standard of dietary treatment in colitis to emphasize that IF is an alternative and still controversial model of nutrition. Point 2 In addition, please add some information on the influence of dietary factors on the modulation of the intestinal microbiota.(e.g. Moszak M, Szulińska M, Bogdański P. You Are What You Eat-The Relationship between Diet, Microbiota, and Metabolic Disorders-A Review. Nutrients. 2020 Apr 15;12(4):1096. doi: 10.3390/nu12041096. PMID: 32326604; PMCID: PMC7230850.) Point 3 The introduction omitted the issue of the impact of IF on clinical parameters in UC - please add a short paragraph summarizing this topic or clearly indicate if there is no data in this subject area. Point 4 Material and Methods Page 2; secondo line in 2.1: „Mice in the SIF group or LIF group were fed every other day (food pellets were provided or removed at 9 am each day) for 2 weeks or 4 months, while mice in the control group were allowed free access to food (SAL or LAL group).” Point 5. The abbreviations SIF and LIF have not been introducted before, so the readable of results may be difficult.Point 6.
In the introduction, weeks (2 or 20) were used to determine the intervention time. I prefer this form instead of “2 weeks or 4 months”.
Results Point 7. The figures and tables contain the most important results but are not fully legible (understandable. Point 8 Because the abbreviations (eg. SIF and LIF) have not been explained before, the results in Figures and Tables are not fully readable. Please add a explanation of abbreviations under tables and figures. Discusion The discussion is well written and based on other relative studies. However, the authors should more highlight the novelty of the present study, the clinical dimension of the study and distinction from previous studies.Minor comments:
Page 3, line 7 -> „p < 0.05” or „p<0.05” (in line 6 notation is without space)
Page 10, line 277 -> the abbreviation COVID-19 was introduced before. Please, use the abbreviation consistently.
Inconsistency in the use of abbreviations - each abbreviation should be explained on first use and then used consistently throughout the manuscript. There is lots of error in this field eg: Abbreviations in Abstract (DSS, KC, DAI) Page 2, line 20 -> “IBD” Page 2, line 22 -> “UC” (have been explained in line 24) Page 3, line 3 -> “SCFA” and “BA”Figures 3-9 -> other fonts than Palatino Linotype
Page 10, line 1 -> without numeration (4.1)?
Author Response
Point 1. In my opinion, the Introduction would be enriched by adding a few sentences informing about the standard of dietary treatment in colitis to emphasize that IF is an alternative and still controversial model of nutrition.
Epidemiological data suggest that vegetarian diet and gluten-free diet failed to have robust evidence on alleviating ulcerative colitis (UC). Then, the low FODMAP (fermentable oligosaccharides, disaccharides, monosaccharides, and polyols) diet (LFD) may be helpful, especially for overlapping functional symptoms in those with inflammatory bowel disease (IBD). However, the data on whether or not LFD impacts IBD disease activity is inconclusive. Fasting/reduced-calorie diets are promising interventions. A study conducted in 2008 found that a significant decrease in a clinical colitis activity index was noted in 60 IBD patients underwent Ramadan.
Point 2 In addition, please add some information on the influence of dietary factors on the modulation of the intestinal microbiota.(e.g. Moszak M, Szulińska M, Bogdański P. You Are What You Eat-The Relationship between Diet, Microbiota, and Metabolic Disorders-A Review. Nutrients. 2020 Apr 15;12(4):1096. doi: 10.3390/nu12041096. PMID: 32326604; PMCID: PMC7230850.)
As we all known, composition of the diet, dietary pattern, and long-term dietary habits shape the diversity and composition of gut microbiome. Studies shown that the α-diversity (richness) and β-diversity (variety) of gut microbiota was impacted by fasting[14]. It is worth mentioning that in our study IF did not change the α-diversity of gut microbiota. According to a study, IF decreased the enrichments of colitis-related microbes such as Shigella and Escherichia and increased the relative abundance of Rikenellaceae, Lactobacillus, Coproccus, and Ruminococcus.
Point 3 The introduction omitted the issue of the impact of IF on clinical parameters in UC - please add a short paragraph summarizing this topic or clearly indicate if there is no data in this subject area.
A research published in the 2022 has demonstrated that IF was associated with worsening of the partial Mayo score. This study has found that serum CRP and stool calprotectin did not show a significant change after IF. On the contrary, another study indicated that IF exhibited the protective effects against colitis and related behavioral disorders[13].
Point 4 Material and Methods Page 2; secondo line in 2.1: „Mice in the SIF group or LIF group were fed every other day (food pellets were provided or removed at 9 am each day) for 2 weeks or 4 months, while mice in the control group were allowed free access to food (SAL or LAL group).” Point 5. The abbreviations SIF and LIF have not been introducted before, so the readable of results may be difficult.
Each group contained eight mice. Mice in the short-term IF group (SIF) or long-term IF group (LIF) were fed every other day (food pellets were provided or removed at 9 am each day) for 2 weeks or 20 weeks
Point 6.
In the introduction, weeks (2 or 20) were used to determine the intervention time. I prefer this form instead of “2 weeks or 4 months”.
Has been corrected.
Results Point 7. The figures and tables contain the most important results but are not fully legible (understandable)
The figures and tables have been corrected.
Point 8 Because the abbreviations (eg. SIF and LIF) have not been explained before, the results in Figures and Tables are not fully readable. Please add a explanation of abbreviations under tables and figures.
Has been corrected.
Discusion The discussion is well written and based on other relative studies. However, the authors should more highlight the novelty of the present study, the clinical dimension of the study and distinction from previous studies.
It is widely accepted that the gut microbiome is a contributor to the development of UC. Several studies reported that IF prompted recovery from colitis in animal models. However, the roles of gut microbiota involved in colitis with IF regimen need to be further investigated. And microbial metabolites, such as short chain fatty acids (SCFAs), secondary bile acids, and tryptophan also plays a crucial role in the pathogenesis of UC. Most related research in this area has focused on the influence of IF on the gut microbiome. Fewer studies have demonstrated the dynamic changes of both microbiome and metabolome. Our results indicated that IF significantly elevated the relative abundances of Muribaculum and Akkermansia in the gut which attenuated DSS-induced colitis. Moreover, the increased concentrations of inosine and secondary bile acids like LCA produced by the gut microorganisms probably contributed highly to the anti-colitis effects of IF.
Minor comments:
Page 3, line 7 -> „p < 0.05” or „p<0.05” (in line 6 notation is without space)
Has been corrected.
Page 10, line 277 -> the abbreviation COVID-19 was introduced before. Please, use the abbreviation consistently.
Inconsistency in the use of abbreviations - each abbreviation should be explained on first use and then used consistently throughout the manuscript. There is lots of error in this field eg: Abbreviations in Abstract (DSS, KC, DAI) Page 2, line 20 -> “IBD” Page 2, line 22 -> “UC” (have been explained in line 24) Page 3, line 3 -> “SCFA” and “BA”
Has been corrected.
Figures 3-9 -> other fonts than Palatino Linotype
Has been corrected.
Page 10, line 1 -> without numeration (4.1)?
Has been corrected.
Reviewer 2 Report
In the present study, authors investigated effects of intermittent fasting on gut microbiota and DSS-induced colitis.
I think this study have adequately conducted and well written.
Information shown in this manuscript can help to understand.
I have several minor comments.
1. “Butyrate acid”, ”Acetate acid”, and “Propionate acid” should be “Butyrate”, Acetate”, and “Propionate”, respectively.
2. In Methods. Supplementary methods should be combined to Methods section.
3. In Methods. Please mention the age (week) of the mouse and sample size of SIF and LIF groups. Please describe the feeding situation (fed or fasted) and time (AM or PM) of the mice dissection.
4. In Methods. Daily intake of DSS must be shown.
5. In Table 1. Alpha diversity of the gut microbiota is quite low in long-term feeding groups. Please discuss that.
6. In Figure 2 and 4. Letters are too small to see.
7. Page 10 line 9. What does BM mean?
8. The title is overstated. There is no data that modulating the gut microbiota and metabolome alleviates colitis in IF group.
Finally thank you for your interesting manuscript. I enjoyed reading it.
Author Response
- “Butyrate acid”, ”Acetate acid”, and “Propionate acid” should be “Butyrate”, Acetate”, and “Propionate”, respectively.
Has been corrected.
- In Methods. Supplementary methods should be combined to Methods section.
Has been corrected.
- In Methods. Please mention the age (week) of the mouse and sample size of SIF and LIF groups. Please describe the feeding situation (fed or fasted) and time (AM or PM) of the mice dissection.
Has been corrected.
- In Methods. Daily intake of DSS must be shown.
Supplementary Materials Table S1. The daily intake of DSS (g/day)
|
|
ALD |
IFD |
||
|
|
Cage 1 (n=4) |
Cage 2 (n=4) |
Cage 3 (n=4) |
Cage 4 (n=4) |
|
Day 1 |
14.30 |
16.22 |
16.45 |
16.25 |
|
Day 2 |
16.29 |
16.75 |
16.36 |
18.04 |
|
Day 3 |
15.81 |
15.01 |
14.84 |
16.44 |
|
Day 4 |
13.06 |
14.44 |
14.13 |
15.94 |
|
Day 5 |
10.89 |
11.86 |
11.83 |
13.16 |
|
Day 6 |
9.78 |
10.73 |
11.95 |
9.86 |
|
Day 7 |
15.96 |
14.82 |
14.28 |
14.78 |
- In Table 1. Alpha diversity of the gut microbiota is quite low in long-term feeding groups. Please discuss that.
The gut microbiota undergoes extensive changes across the lifespan. In this part, we just compared the influence of short term IF on the gut microbiota between SAL and SIF group, and long term IF on the gut microbiota between LAL and LIF group.
- In Figure 2 and 4. Letters are too small to see.
Has been corrected.
- Page 10 line 9. What does BM mean?
On the other hand, the cumulative food consumption of the four groups was calculated, and the food consumption of the IF group was 295.78 g/100 g Body mass (BM), while that of the AL group was 277.21 g/100 g BM (Figure 5 B), the IFD group was 292.14 g/100 g BM, and the ALD group was 269.16 g/100 g BM.
- The title is overstated. There is no data that modulating the gut microbiota and metabolome alleviates colitis in IF group.
It has been found that the composition of gut microbes and microbial metabolites are enormously altered after IF. The abundance of Bacteroides, Muribaculum and Akkermansia was prominently elevated after IF which were probably associated with metabolic benefits on colitis according to the correlation analysis. Moreover, correlation analysis shown that microbial metabolites like LCA and inosine produced in the gut drove a positive role in colitis. So, IF maybe have the ability to modulate the gut microbiota and metabolome to alleviate colitis.